# SARS-CoV-2 Antinucleocapsid Antibody Response of mRNA and Inactivated Virus Vaccines Compared to Unvaccinated Individuals

**DOI:** 10.3390/vaccines10050643

**Published:** 2022-04-20

**Authors:** Arwa Qaqish, Manal Mohammad Abbas, Mohammad Al-Tamimi, Manal Ahmad Abbas, Mariam Al-Omari, Rami Alqassieh

**Affiliations:** 1Department of Biology and Biotechnology, The Hashemite University, Zarqa 13133, Jordan; arwa@hu.edu.jo; 2Department of Medical Laboratory Sciences, Faculty of Allied Medical Sciences, Al-Ahliyya Amman University, Amman 19328, Jordan; m.abbas2@ammanu.edu.jo (M.M.A.); m.abbas@ammanu.edu.jo (M.A.A.); 3Pharmacological and Diagnostic Research Lab, Al-Ahliyya Amman University, Amman 19328, Jordan; 4Department of Basic Medical Sciences, Faculty of Medicine, The Hashemite University, Zarqa 13133, Jordan; 5Faculty of Medicine, Yarmouk University, Irbid 21163, Jordan; maryam.o@yu.edu.jo; 6Department of General and Specialized Surgery, Faculty of Medicine, The Hashemite University, Zarqa 13133, Jordan; rami_qaisieh@hu.edu.jo

**Keywords:** SARS-CoV-2, COVID-19, anti-N, anti-S, IgG, IgM, Pfizer, Sinopharm

## Abstract

Comparative studies of SARS-CoV-2 antinucleocapsid (anti-N) antibody response in the context of inactivated virus vaccines versus natural infection are limited. This study aims to determine and compare the anti-N antibody levels in people vaccinated with Sinopharm’s (Wuhan, China) inactivated virus vaccine in comparison with naturally infected unvaccinated and Pfizer’s spike (S) mRNA-based vaccinated subjects. Two hundred ninety-nine Jordanian adults participated in the study including unvaccinated COVID-19-infected patients (*n* = 99), Pfizer-vaccinated (*n* = 100), and Sinopharm-vaccinated recipients (*n* = 100). Serum samples were assayed for anti-N IgG, anti-N IgM, and anti-S IgG. Sera of 64.6% of naturally infected unvaccinated participants had positive anti-S IgG (median = 36.35 U/mL; range: 0.04–532.5 U/mL) compared to 88% of Pfizer-vaccinated (Manhattan, NY, USA) (median = 26.52 U/mL; range: 0.39–1265 U/mL) and 58% of Sinopharm-vaccinated subjects (median = 14.35 U/mL; range: 0.39–870.17 U/mL). Samples of 60.6% of naturally infected unvaccinated people had positive anti-N IgG (median = 15.03 U/mL; range: 0–265.1 U/mL) compared to 25% of Pfizer-vaccinated (median = 0.02 U/mL; range: 0–68 U/mL) and 48% of Sinopharm-vaccinated subjects (median = 0.8 U/mL; range: 0–146.3 U/mL). Anti-N titers among the three groups were significantly different (*p* < 0.05). Anti-N IgM antibodies appeared in 23.2% of the naturally infected unvaccinated group (median = 0.29 U/mL; range: 0–15 U/mL) compared to only 9.0% of Pfizer-vaccinated (median = 018 U/mL; range: 0–33 U/mL) and 7.0% of Sinopharm-vaccinated subjects (median = 0.2 U/mL; range: 0–12.02 U/mL). A significant negative correlation was found between anti-S and age for both vaccines and between anti-S and the presence of chronic disease in Sinopharm-vaccinated subjects. A significant positive correlation between anti-N and anti-S titers was found among the three groups. This study shows that the inactivated virus vaccine, Sinopharm, induces an anti-N response that can boost that of natural infection or vice versa. On the other hand, the Pfizer mRNA-based vaccine induces a significantly stronger anti-S Ab response.

## 1. Introduction

Since the coronavirus disease 2019 (COVID-19) was declared a pandemic, there was no doubt that the only way to control the explosive spread of the severe acute respiratory syndrome coronavirus 2 (SARS-CoV-2) was by inventing a safe and protective vaccine [1,2]. An effective vaccine would prevent the virus from entering host cells in the forthcoming months and/or years in the context of infection by evoking the production of virus-specific neutralizing antibodies (Abs) and long-lived memory B cells, respectively [3,4,5].

Among >250 COVID-19 vaccine candidates being designed and formulated at different stages of development around the world, BNT162 (Pfizer–BioNTech) was the first to be approved for use worldwide by the WHO [6,7,8,9]. Pfizer is an mRNA-based vaccine and a lipid nanoparticle formula harboring nucleoside-modified RNA encoding the SARS-CoV-2 full-length spike [8,9,10]. Another vaccine that predominated in many Asian and developing countries is Sinopharm/BBIBP-CorV [11]. Sinopharm is a classical inactivated virus-based vaccine derived from vero-cells [8,9,12].

The 30 Kb positive-strand RNA genome of SARS-CoV-2 encodes four essential structural proteins and several smaller “accessory” proteins. The spike (S) protein, nucleocapsid (N) protein, membrane (M) protein, and the envelope (E) protein are all required to produce a structurally complete viral particle [13]. Among these proteins, S and N proved to be the most immunogenic [14] as they evoke the production of a strong Ab response in variable hosts [15].

The S protein is of special importance since it mediates attachment of the virus to the host cell surface receptor(s) and subsequent viral entry into the host cell [16]. Hence, it is the most targeted protein for the development of COVID-19 therapeutic Abs and vaccines [17,18].

N is the only protein that functions primarily to bind to the SARS-CoV-2 RNA genome, making up the nucleocapsid. Although N is largely involved in processes related to the viral genome, it is also involved in other aspects of the SARS-CoV-2 replication cycle, namely viral assembly, budding, and the host cellular response to viral infection [19].

Since pathogen-specific Abs often point to immunological mechanisms of protection, many studies were conducted to understand why some individuals recover from infection, whereas others rapidly progress to pneumonial complications that may lead to death. It has been suggested that early antigen-specific humoral responses to SARS-CoV-2 could yield differences in disease prognosis [20]. Interestingly, it has been reported that patients who recovered from the disease generated anti-S neutralizing Abs in higher concentrations compared to patients with serious symptoms who generated higher anti-N Abs [20,21]. 

Whether inactivated virus-based vaccines induce the production of anti-N Ab responses in vaccinated people is weakly investigated. Among a few studies, some reported the absence of anti-N Abs in the sera of Sinopharm vaccinees [22], and some reported detectable levels of such Abs in less than 50% of the cases [23].

Most COVID-19 vaccines were designed to induce the production of anti-S neutralizing Abs that would prevent viral entry and facilitate virus elimination. Comparing the efficacy of the Pfizer vaccine with that of Sinopharm showed a potential higher protection rate for Pfizer correlating with a higher percentage of seroconverted vaccinees and higher titers of neutralizing anti-S Abs [22,23,24]. Moreover, the Pfizer vaccine showed an approximate 95% efficacy rate in protection against subsequent infections with SARS-CoV-2 in terms of symptom severity and need for hospitalization [10]. On the other hand, it has been reported that Sinopharm vaccine recipients had a higher risk of postvaccination infections, hospitalizations, ICU admissions, and deaths than Pfizer–BioNTech recipients [25].

This study aims to determine the anti-N Ab response in people vaccinated with the inactivated SARS-CoV-2 virus, Sinopharm, in comparison with naturally infected unvaccinated and Pfizer S protein mRNA-vaccinated subjects.

## 2. Materials and Methods

### 2.1. Participants, Setting, and Ethical Consideration

The present cross-sectional study was conducted between January 2021 and June 2021. Two hundred ninety-nine Jordanian adults participated voluntarily in the study. Study population was recruited at Prince Hamza Hospital (PHH) and was divided into 3 cohorts: confirmed previous natural infected COVID-19 patients, Sinopharm-vaccinated group, and Pfizer–BioNTech-vaccinated group. Vaccinated groups received two doses of either Sinopharm or Pfizer–BioNTech COVID-19 vaccines, while the COVID-19 natural infection group was confirmed by a documented positive RT-PCR test using a nasopharyngeal swab and performed by an accredited laboratory. The following participants were excluded from the study: patients with history of allergies or anaphylaxis, immunocompromised patients, and patients on corticosteroids or immunosuppressing medications. For each participant, the demographic, clinical, and social data were collected. The study was conducted according to the guidelines of the Declaration of Helsinki and was approved by the institutional review board (IRB) committee at the Hashemite University (No. 88 6/7/2020/2021) and PHH. 

### 2.2. Demographic and Clinical Characteristics of Population Study

A total of 299 participants were recruited in the study, including COVID-19-confirmed patients (*n* = 99) up to six months post-infection with documented positive COVID-19 RT-PCR test, Pfizer–BioNTech-vaccinated recipients (*n* = 100), and Sinopharm-vaccinated recipients (*n* = 100) from PHH. Demographic and clinical data of the 3 studied groups are illustrated in Table 1.

### 2.3. Sample Collection

Serum samples were collected from participants at PHH after signing a consent form. The naturally infected COVID-19 samples were collected up to 6 months after a confirmed RT-PCR test. Vaccinated group samples were collected 2 weeks after the second vaccination. 

### 2.4. Sample Analysis

All serum samples were analyzed using the Mindray anti-SARS-CoV-2 S assay on the CL-series Chemiluminescence Immunoassay Analyzer (Mindray Diagnostics) for the quantitative detection of antibodies to the SARS-CoV-2 spike protein receptor binding domain (RBD) [26,27]. Samples with a concentration of <10.00 U/mL were considered negative. Calibrations were done before running the test, and controls were run simultaneously. Same serum samples were analyzed using COVID-19 IgG/IgM Duo for quantitative detection of anti-coronavirus Nucleocapsid IgG and IgM in human serum by fluorescence immunoassay (FIA) using the FREND System. The protocol was run according to company procedure (NanoEntek) [28]. Samples with a concentration of <1.00 U/mL were considered negative.

### 2.5. Statistical Analysis

For statistical analysis, the Statistical Package for the Social Sciences (SPSS) version 22.0 (Chicago, IL, USA) was used [29]. Categorical variables were summarized as frequencies and percentages, whereas numerical variables were represented by the mean (±SD). The normality of distribution of IgG anti-N and anti-S titers was checked by both the Kolmogorov–Smirnov and Shapiro–Wilk tests. The difference in medians of the positive anti-N IgG titers (after excluding the negative results) for the 3 groups and the difference in medians of the positive anti-S IgG titers for the 3 groups were carried out using an independent-sample Kruskal–Wallis test, since data were not normally distributed, followed by Dunn’s post hoc test. *p* ≤ 0.05 was considered significant. Mann–Whitney U test was utilized to compare anti-N and anti-S in Pfizer–BioNTech-vaccinated, Sinopharm-vaccinated, and COVID-19 infected groups. Spearman’s correlation coefficient was obtained after 2-tailed bivariate correlation analysis of each antibody titer vs. age, gender, chronic diseases, smoking status, or previous infection with COVID-19. On the other hand, Person’s correlation coefficient was obtained after 2-tailed bivariate correlation analysis of anti-N vs. anti-S titers. In all statistical analysis tests, *p* ≤ 0.05 was considered as significant. Figures were generated using the GraphPad Prism version 8.0.0 for Windows, the GraphPad Software, San Diego, CA, USA, www.graphpad.com (accessed on 27 February 2022) [30].

## 3. Results

### 3.1. Comparing Anti-S IgG Levels in Naturally Infected Unvaccinated and Vaccinated Subjects

Sera of 64.6% (66/99) of naturally infected unvaccinated participants had positive anti-S Abs compared to 88% (88/100) of Pfizer-vaccinated and 58% (58/100) of Sinopharm-vaccinated subjects. No statistically significant difference among the three groups was found in the level of anti-S titers (*p* > 0.05) (Table 2, Figure 1A). However, when only positive anti-S samples were considered, titers of naturally infected unvaccinated people (median = 36.35 U/mL; range: 0.04–532.5 U/mL) were significantly higher compared to titers of the Sinopharm-vaccinated group (median = 26.52 U/mL; range: 0.39–1265 U/mL) (*p* < 0.002). Similarly, naturally infected unvaccinated sera showed significantly higher anti-S titers than Pfizer-vaccinated sera (median = 14.35 U/mL; range: 0.39–870.17 U/mL) (*p* < 0.0001). On the other hand, no significant difference was found between titers of Pfizer-vaccinated and Sinopharm-vaccinated individuals (*p* > 0.05) (Table 2, Figure 1B).

### 3.2. Comparing Anti-N IgG Levels in Naturally Infected Unvaccinated and Vaccinated Subjects

The sera of 60.6% (60/99) of naturally infected unvaccinated participants had positive anti-N Abs compared to 25% (25/100) of Pfizer-vaccinated and 48% (48/100) of Sinopharm-vaccinated subjects. The anti-N IgG titers of naturally infected unvaccinated sera (median = 15.03 U/mL; range: 0–265.1 U/mL) were significantly higher compared to titers of Pfizer-vaccinated people (median = 0.02 U/mL; range: 0–68 U/mL) (*p* < 0.0001). Similarly, Sinopharm-vaccinated anti-N titers (median = 0.8 U/mL; range: 0–146.3 U/mL) were significantly higher compared to titers of the Pfizer-vaccinated group (*p* < 0.008) (Table 2, Figure 2A). On the other hand, no significant difference was found between the titers of naturally infected unvaccinated and Sinopharm-vaccinated groups (*p* > 0.05). When only seropositive samples were considered, the anti-N titers of the naturally infected unvaccinated sera were significantly higher than those of the Sinopharm-vaccinated group (*p* < 0.001). No significant difference was found between the anti-N levels (*p* > 0.05) of the Pfizer and Sinopharm-vaccinated groups. Similarly, the anti-N levels of naturally infected unvaccinated and Pfizer-vaccinated groups were not significantly different (*p* > 0.05) (Table 2, Figure 2B). 

### 3.3. Comparing Anti-N IgM Antibody Levels in Naturally Infected Unvaccinated Group and Vaccinated Subjects

In the naturally infected unvaccinated group, 23.2% (23/99) (median = 0.29 U/mL; range: 0–15 U/mL) had positive anti-N IgM compared to only 9.0% (9/100) (median = 018 U/mL; range: 0–33 U/mL) of Pfizer-vaccinated subjects and 7.0% (7/100) (median = 0.2 U/mL; range: 0–12.02 U/mL) of Sinopharm-vaccinated subjects (Table 2, Figure 3A). The anti-N IgM titers of the naturally infected unvaccinated participants were significantly higher compared to those of the Pfizer-vaccinated group (*p* < 0.026) but not the Sinopharm-vaccinated group (*p* > 0.05). Considering the seropositive samples only, no significant difference was found among the anti-N IgM levels of the three groups (*p* > 0.05) (Table 2, Figure 3B).

### 3.4. Anti-N and Anti-S IgG Titers of Anti-N IgM Positive Samples in the 3 Study Groups

The anti-N IgM Abs were detected in 23 naturally infected unvaccinated subjects. The median IgM titer was 2.1 (IQR: 3.08) for this group, while the median titer of anti-N IgG for the same individuals was 35.75, and the median titer of anti-S IgG was 189.05. Positive IgM was found in nine subjects vaccinated with the Pfizer vaccine. For this group, the median IgM titer was 1.93 (IQR: 3.16), while the median anti-N IgG titer was 0.14, and the median anti-S IgG titer was 21. On the other hand, anti-N IgM Abs were found in seven subjects vaccinated with the Sinopharm vaccine. Here, the median IgM titer was 1.4 (IQR: 9.22), while the median anti-N IgG was 21.47, and the median of anti-S IgG was 189.19 (Table 3).

### 3.5. Effect of Age, Gender, Chronic Diseases, Smoking, Previous Infection, Department Admission, and Symptom Severity on Anti-S and Anti-N Ab Levels in the 3 Study Groups

Pertaining to the anti-S IgG levels, a significant weak negative correlation was found between anti-S titers and age for both vaccines and a significant positive correlation with natural infection (Table 4). Similarly, a negative correlation was found between anti-S and the presence of chronic disease in Sinopharm-vaccinated subjects (Table 4). On the other hand, no significant correlation was found with other variables (Table 4). 

No correlation was found between anti-N titers and age, gender, presence of chronic diseases, smoking status, or previous COVID-19 infection in the three study groups (Table 4 and Table 5). A significant positive correlation between anti-N and anti-S titers was found in all three groups. A significant positive weak correlation was found between anti-N levels and department, as outpatients showed higher Ab responses compared to inpatients. Such a significantly positive weak correlation was also found between anti-N levels and the time of sample collection after a positive RT-PCR confirmation of COVID-19 but not with symptoms or their severity. Similar results were obtained for anti-S levels (Table 5).

## 4. Discussion

The purpose of vaccination against SARS-CoV-2 is to induce the production of neutralizing Abs that would prevent viral entry and facilitate virus elimination. Still, many COVID-19-associated deaths and complicated cases of acute respiratory distress syndrome (ARDS) correlated with very high Ab titers, specifically against the N protein [21]. The anti-N Ab response is not very well understood. It has been reported that one fifth of the cases naturally infected with SARS-CoV2 does not elicit a measurable anti-N response [31]. It has also been reported that vaccinated people with inactivated virus-based vaccines do not show anti-N Abs in their sera [22]. In this investigation, we aimed to study the anti-S and anti-N-Ab response in naturally infected patients, patients vaccinated with Pfizer’s S protein mRNA-based vaccine, and patients vaccinated with Sinopharm’s inactivated virus-based vaccine. 

Looking at the anti-S Ab response, 64.6% (64/99) of naturally infected patients showed anti-S Abs in their sera. This percent could be low taking into consideration that all patients were PCR-confirmed to be infected with SARS-CoV-2. We believe the reason might be the timing of the blood collection, as 6 months post symptom onset could lead to the disappearance of anti-COVID-19-specific Abs in asymptomatic patients and those with low viral loads [32,33]. Considering anti-S positive samples, Ab titers were variable from low to very high (range: 0.39–1265). This variation correlated negatively with the presence of chronic diseases and with age. A few other studies reported increased Ab titers with increased age [34,35]. More severe symptoms, delayed Ab response, and higher viral replication could be the reason why older patients show higher Ab titers in their sera compared to younger ones, especially at longer times (6 months) after infection. 

In terms of symptoms, previous studies found that anti-S Ab titers also correlate highly with symptom severity and negatively with viral loads [36]. A significant positive weak correlation was found between anti-S as well as anti-N levels and department, as naturally infected unvaccinated outpatients showed higher Ab responses compared to inpatients (Table 5). We believe that this could be due to differences in the timing of blood collection. Samples were collected during hospitalization within the first 2 weeks post onset of COVID-19 symptoms from inpatients, whereas outpatients donated their samples up to 6 months after. Another reason could be a boosting effect due to potential multiple infections with SARS-CoV-2. In fact, such a reason is supported by the presence of anti-N IgM antibodies in sera of patients 6 months after PCR testing.

The Pfizer-vaccinated group showed a higher positivity rate (88%) (88/100) compared to the Sinopharm-vaccinated group (58%) (58/100) with significantly higher anti-S titers (*p* < 0.05). We are not the first to report that the Pfizer vaccine has more efficacy in developing protective anti-S Abs compared to Sinopharm [22,23,24]. A totally pure S protein-producing vaccine (Pfizer) must produce higher anti-S antibody titers compared to an inactivated whole virus harboring all different kinds of viral antigens (Sinopharm). Interestingly, taking into consideration positive cases (cases showing detectable titers of anti-S IgG), the titers of the Sinopharm-vaccinated group were significantly higher than those of the Pfizer-vaccinated group (*p* < 0.05). Still, the reason why some people do not show a detectable Ab response to vaccines, especially to the extent seen in the Sinopharm group of our study, is an immunological mystery that should be investigated.

In this study, although the majority of Pfizer and Sinopharm-vaccinated groups did not have previous infection with SARS-CoV-2 (Table 1), 25% (25/100) and 48% (48/100) of participants showed anti-N Abs in their sera, respectively. Samples from these groups were collected at the beginning of the vaccination campaign driven by the ministry of health in Jordan, when people showing a record of natural infection were not allowed to take any vaccines. Since the Pfizer vaccine itself does not evoke an anti-N response, these 25% anti-N positive cases must have been previously infected with COVID-19 or were experiencing a current infection, as shown by the anti-N IgM results. Interestingly, anti-N Abs appeared in 48% (48/100) of the Sinopharm group, a significantly higher percentage compared to the Pfizer group. Although previous studies have shown that vaccinated people with inactivated virus-based vaccines do not show measurable anti-N Abs in their sera [22], it could be speculated that the inactivated virus in the Sinopharm vaccine normally elicits a weak and undetectable anti-N Ab response that was boosted by a current natural infection or that the vaccine itself serves as a booster to a previous mild or asymptomatic infection. 

Populations around the globe are subjected to and experiencing multiple infections with the emerging variants of the nonstop SARS-CoV-2 spread. In such case, Sinopharm could amplify the production of anti-N antibodies. Similarly, being vaccinated with inactivated virus-based vaccines produces an anti-N response that could be boosted by subsequent natural infection. Whether these high levels of anti-N Abs will serve to enhance the protective effect or increase the chances of the development of antibody-dependent enhancement (ADE) complications require further investigations.

ADE is a mechanism that is well-defined in viral infections, where non-neutralizing Abs bound to the virus serve as keys that enable viral access to host immune cells [37,38]. ADE has been observed in many viral infections, mainly with positive-strand RNA viruses such as dengue virus, influenza virus, human immunodeficiency virus-1 (HIV-1), West Nile virus, and MERS [39]. The mechanism of ADE in the context of COVID-19 is still under investigation. So far, macrophages infected with the virus proved to play an essential role in viral spread, excessive production of proinflammatory cytokines, and activation-induced lymphocyte death leading to higher viremia and severe lymphopenia [40,41,42,43,44,45,46]. 

The N protein of SARS-CoV-2 has high homology with other highly pathogenic members of the coronavirus family, with a molecular weight of about 46-kDa. It has been reported that previous exposure to non-COVID-19 human coronaviruses could lead to the production of anti-N non-neutralizing Abs and establish an overwhelming proinflammatory state through ADE [21]. In COVID-19, many have reported a strong correlation between anti-N Ab titers and the need for critical care [20,21]. 

A vaccine efficacy evaluation reported that Sinopharm vaccine recipients had a higher risk of postvaccination infections, hospitalizations, ICU admissions, and deaths compared to Pfizer–BioNTech recipients [25]. Could the Sinopharm-amplified anti-N response and potential non-neutralizing anti-N-driven AED complications be reasons behind this observed lower protective efficacy? Further investigation is needed.

The major limitation of our study is that we did not measure total non-neutralizing IgG and non-neutralizing anti-S and anti-N proteins. In addition, the number of samples per group is low, and the timing of the sample collection was not consistent among groups: 6 months post symptom onset for the naturally infected group and 2 weeks post full vaccination for the vaccinated group. 

## 5. Conclusions

In conclusion, this study shows that the inactivated virus vaccine, Sinopharm, induces an anti-N response that can boost that of natural infection or vice versa. On the other hand, the Pfizer mRNA-based vaccine induces a significantly stronger anti-S Ab response. Many factors could affect anti-S and anti-N Ab levels among vaccinated individuals including age, chronic diseases, and duration of sampling after vaccination. Anti-N Ab detection could be a helpful tool to survey past and current COVID-19 infections among a vaccinated population with mRNA-based vaccines. The roles of high levels of anti-N Abs after vaccination, natural infection, or both in the development of protective immunity versus ADE-related complications need further investigation.

## Figures and Tables

**Figure 1 vaccines-10-00643-f001:**
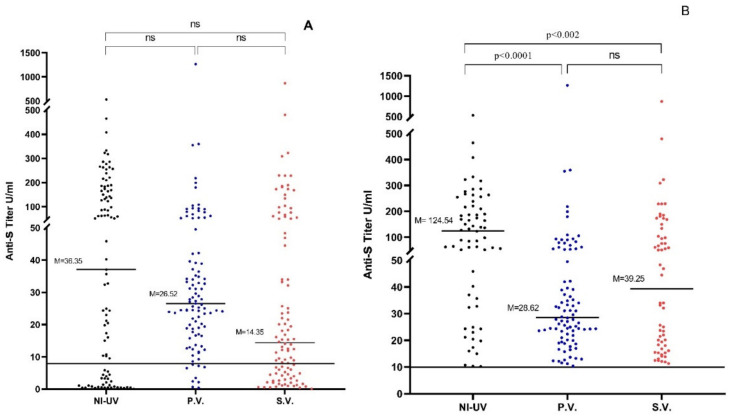
(**A**) Anti-S IgG titers in the 3 studied groups for all participants. (**B**) Anti-S IgG titers in the 3 studied groups for all participants after excluding the seronegative samples. NI-UV: Naturally Infected Unvaccinated, P.V.: Pfizer-Vaccinated, S.V.: Sinopharm-Vaccinated, M: Median, ns: Nonsignificant (*p* > 0.05).

**Figure 2 vaccines-10-00643-f002:**
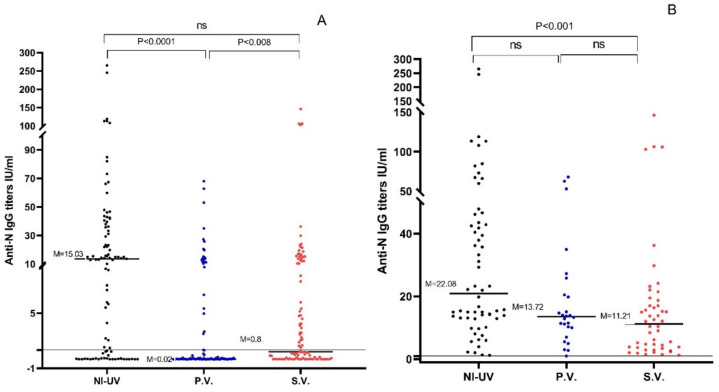
(**A**) Anti-N IgG titers in the 3 studied groups for all participants. (**B**) Anti-N IgG titers in the 3 studied groups after excluding seronegative samples. NI-UV: Naturally Infected Unvaccinated, P.V.: Pfizer Vaccinated, S.V.: Sinopharm Vaccinated, M: Median, ns: Nonsignificant (*p* > 0.05).

**Figure 3 vaccines-10-00643-f003:**
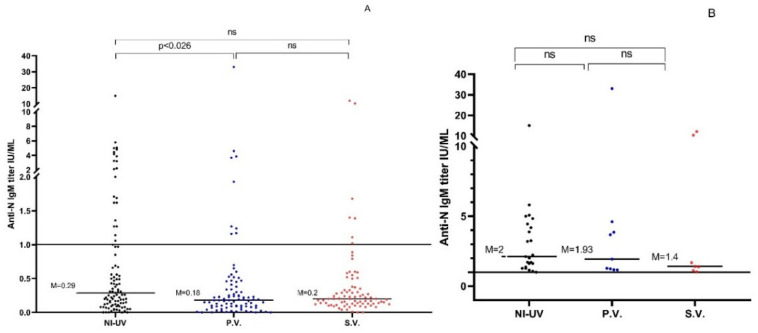
(**A**) Anti-N IgM titers in the 3 studied groups for all participants. (**B**): Anti-N IgM titers in the 3 studied groups after excluding seronegative samples. NI-UV: Naturally Infected Unvaccinated, P.V.: Pfizer Vaccinated, S.V.: Sinopharm Vaccinated, M: Median, ns: Nonsignificant (*p* > 0.05).

**Table 1 vaccines-10-00643-t001:** Demographic and clinical data of COVID-19 patients (*n* = 99), Pfizer-vaccinated recipients (*n* = 100), and Sinopharm-vaccinated recipients (*n* = 100).

	Variable	Naturally Infected Unvaccinated Number (%)	Pfizer-Vaccinated Number (%)	Sinopharm-Vaccinated Number (%)
**Number of participants**		99	100	100
**Age (Years)**	21–40	53 (53.5)	13 (13.3)	9 (9)
	41–60	35 (35.3)	31 (31.6)	31(31)
	61–80	8 (8.1)	54 (55.1)	55 (55)
	NA	3	2	5
**Gender**	Male	57 (57.5)	69 (69)	68 (68)
	Female	42 (42.4)	31 (31)	32 (32)
**Chronic disease**	Yes	NA	57 (58.1)	60 (60)
	No	NA	41 (41.8)	35 (35)
	NA	NA	2	5
**Smoking**	Yes No	NA	63 (64.2)	30 (30)
		NA	31 (31.6)	60 (60)
		NA	2	10
**Previous confirmed infection with COVID-19**	Yes	N-AP	1 (1.0)	4 (4)
	No	N-AP	97 (98.9)	91 (91)
	NA	N-AP	2	5

NA: not available. N-AP: not applicable.

**Table 2 vaccines-10-00643-t002:** Comparing anti-S and anti-N levels in naturally infected unvaccinated and vaccinated subjects.

	Naturally Infected Unvaccinated	Pfizer-Vaccinated	Sinopharm-Vaccinated
**Anti-S IgG titer U/mL**			
% Positive	64.6%	88%	58%
All participants‘ median (IQR)	36.35 (164.51)	26.52 (24.25)	14.35 (46.02)
Median number out of 100	31	45	39
Range	0.04–532.5	0.39–1265	0.39–870.17
Only positive cases median (IQR)	124.54 (183.30)	28.62 (29.74)	39.25 (119.95)
**Anti-N IgG titer U/mL**			
% Positive	60.6%	25%	48%
All participants‘ median (IQR)	15.03 (32.21)	0.02 (2.31)	0.8 (10.41)
Median number out of 100	66	47	50
Range	0–265.1	0–68	0–146.3
Only positive cases median (IQR)	22.08 (32.64)	13.72 (12.59)	11.21 (14.6)
**Anti-N IgM titer U/mL**			
% Positive	23.2%	9.0%	7.0%
All participants‘ median (IQR)	0.29 (0.86)	0.18 (0.3)	0.2 (0.23)
Median number out of 100	49	43	40
Range	0–15	0–33	0–12.02
Only positive cases median (IQR)	2.1 (3.08)	1.93 (3.16)	1.4 (9.22)

IQR: Interquartile range.

**Table 3 vaccines-10-00643-t003:** Comparison among antibody titer median of all three groups for anti-N and anti-S IgG, which are anti-N IgM positive.

	Medians of Anti-S and Anti-N IgG Titers for Anti IgM +ve Samples		
	Naturally Infected Unvaccinated (23 Samples)	Pfizer-Vaccinated (9 Samples)	Sinopharm-Vaccinated (7 Samples)
**Anti-S IgG**	189.06	21	189.19
**Anti-N IgG**	35.75	0.14	21.47

**Table 4 vaccines-10-00643-t004:** Correlations between antibody titers and age, gender, presence of chronic diseases, smoking status, and previous COVID-19 infection.

		Age (Years)	Gender	Chronic Disease	Smoking Status	Previous COVID-19 Infection
**Anti-N IgG for Pfizer-vaccinated group**	Correlation Coefficient	0.022	0.016	0.050	0.165	0.108
	Sig. (2-tailed)	0.832	0.879	0.631	0.113	0.301
**Anti-S IgG for Pfizer-vaccinated group**	Correlation Coefficient	−0.214 *	0.078	−0.156	−0.163	0.131
	Sig. (2-tailed)	0.035	0.444	0.127	0.110	0.200
**Anti-N IgG for Sinopharm-vaccinated group**	Correlation Coefficient	−0.152	−0.015	−0.167	0.085	0.101
	Sig. (2-tailed)	0.142	0.883	0.109	0.418	0.333
**Anti-S IgG for Sinopharm-vaccinated group**	Correlation Coefficient	−0.270 **	0.111	−0.272 **	−0.098	0.185
	Sig. (2-tailed)	0.008	0.0	0.008	0.348	0.072
**Anti-N IgG for naturally infected unvaccinated subjects**	Correlation Coefficient	0.174	0.035	NA	NA	NA
	Sig. (2-tailed)	0.096	0.735	NA	NA	NA
**Anti-S IgG for naturally infected unvaccinated subjects**	Correlation Coefficient	0.216 *	0.087	NA	NA	NA
	Sig. (2-tailed)	0.040	0.407	NA	NA	NA

* Correlation is significant at the *p* < 0.05 level (2-tailed). ** Correlation is significant at the *p* < 0.01 level (2-tailed). NA: not available. Sig: significant.

**Table 5 vaccines-10-00643-t005:** Correlations between antibody titers and symptoms, department, duration, and severity of symptoms in the naturally infected unvaccinated group.

		Symptoms	Department ***	Duration	Severity
**Anti-N**	Correlation Coefficient	0.050	0.246 *	0.292 **	−0.035
	Sig. (2-tailed)	0.641	0.019	0.006	0.739
**Anti-S**	Correlation Coefficient	−0.043	0.296 **	0.301 **	−0.090
	Sig. (2-tailed)	0.683	0.004	0.004	0.391

* Correlation is significant at the *p* < 0.05 level (2-tailed). ** Correlation is significant at the *p* < 0.01 level (2-tailed). NA: not available. Sig: significant. *** Department: Inpatient or Outpatient.

## Data Availability

Data are available upon request.

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
