# Peer review of "SARS-CoV-2 Antinucleocapsid Antibody Response of mRNA and Inactivated Virus Vaccines Compared to Unvaccinated Individuals"

_vaccines, 2022, doi:10.3390/vaccines10050643_

Round 1
Reviewer 1 Report
In the manuscript, Qaqish et al aim to determine and compare the anti-N antibody IgG and IgM levels in humans vaccinated with Sinopharm vaccine and Pfizer vaccine and infected naturally by SARS-CoV-2. The authors observed that the highest rate of anti-N antibody IgG was detected in humans infected naturally, then in humans vaccinated with Sinopharm inactivated vaccine against SARS-CoV-2. Additionally, the authors also detected the highest rate of anti-N antibody IgM in the naturally infected group, then in the group vaccinated with the BNT162 mRNA vaccine. The authors concluded that natural infection and inactivated virus vaccine evoke a weak anti-N Ab response compared to mRNA-based vaccines. The paper should improve in figure presentation, and writing.
- According to the results presented in the manuscript, the conclusion should be that natural infection and inactivated virus vaccine evoked a stronger anti-N Ab response than mRNA-based vaccine, while the later induced a stronger anti-S Ab response compared to the both formers.
- In the discussion, the authors questioned the risk of inactivated vaccine on the long term based on anti-N Abs. This discussion has to improve. Because natural infection induced a higher anti-N response than inactivated vaccine.
- The author should list viruses that ADE is considered as a key mechanism for severe infections.
- What is the relationship between anti-N and human age, gender, etc?
- Due to the prevalence of SARS-CoV-2 around the world and a huge number of asymptomatic infections, it is not easily to differentiate existing antibodies induced by natural infection from vaccination. So the authors carefully make a conclusion.
Author Response
Reviewer 1:
In the manuscript, Qaqish et al aim to determine and compare the anti-N antibody IgG and IgM levels in humans vaccinated with Sinopharm vaccine and Pfizer vaccine and infected naturally by SARS-CoV-2. The authors observed that the highest rate of anti-N antibody IgG was detected in humans infected naturally, then in humans vaccinated with Sinopharm inactivated vaccine against SARS-CoV-2. Additionally, the authors also detected the highest rate of anti-N antibody IgM in the naturally infected group, then in the group vaccinated with the BNT162 mRNA vaccine. The authors concluded that natural infection and inactivated virus vaccine evoke a weak anti-N Ab response compared to mRNA-based vaccines. The paper should improve in figure presentation, and writing.
- According to the results presented in the manuscript, the conclusion should be that natural infection and inactivated virus vaccine evoked a stronger anti-N Ab response than mRNA-based vaccine, while the later induced a stronger anti-S Ab response compared to the both formers.
Response: thank you. We agree, this is correct. Both natural infection and inactivated virus Sinopharm vaccine induce anti-N Ab response whereas Pfizer mRNA-based vaccine induce a strong anti-S response.
Pfizer carry instructions for the synthesis of the spike (S) protein only. So, recipients of such vaccines are supposed to produce anti-S antibodies only and never anti-nucleocapsid (N) antibodies. The appearance of anti-N antibodies in 25% of sera of Pfizer vaccinated subjects must have resulted from a previous or current infection with SARS-CoV-2.
In our study, samples from this group were collected at the beginning of the vaccination campaign driven by the ministry of health in Jordan, when personnel showing a record of natural infection were not allowed to take any vaccines. Because of that we assumed that our study subjects might had either a previous asymptomatic infection, a non-recorded infection or a current infection (during the course of vaccination).
Because of the above, we couldn’t say that Pfizer vaccine evokes an anti-N response.
This text was modified in abstract and conclusion to read This study shows that the inactivated virus vaccine, Sinopharm, induces an anti-N response that can boost that of natural infection or vice versa. On the other hand, the Pfizer mRNA-based vaccine induces a significantly stronger anti-S Ab response.
- In the discussion, the authors questioned the risk of inactivated vaccine on the long term based on anti-N Abs. This discussion has to improve. Because natural infection induced a higher anti-N response than inactivated vaccine.
Response: thank you. The relevant section in discussion was improved as follow:
the following text was deleted If this is the case, the question remains whether an inactivated-virus based vaccine, such as Sinopharm, could be safe on the long term
The following text was modified Populations around the globe are subjected to and experiencing multiple infections with the emerging variants of the non-stopping SARS-CoV-2 spread. In such case, Sinopharm could amplify the production of anti-N antibodies. Similarly, being vaccinated with inactivated virus-based vaccines produces an anti-N response that could be boosted by subsequent natural infection. Whether these high levels of anti-N Abs will serve to enhance the protective effect, or it will increase the chances of development of antibody de-pendent enhancement (ADE) complications require further investigations.
- The author should list viruses that ADE is considered as a key mechanism for severe infections.
Response: Modification done as advised.
- What is the relationship between anti-N and human age, gender, etc?
Response: Modification done as advised.
No correlation was found between anti-N titers and age, gender, presence of chronic diseases, smoking status or previous COVID-19 infection in the 3 study groups.
- Due to the prevalence of SARS-CoV-2 around the world and a huge number of asymptomatic infections, it is not easily to differentiate existing antibodies induced by natural infection from vaccination. So the authors carefully make a conclusion.
Response: Thank you.
Anti-N Ab detection could be a helpful detection tool to survey COVID-19 past and current infections among a vaccinated population with mRNA-based vaccines but not inactivated virus-based vaccines. However, anti-N IgM testing could reflect the percent of population currently infected with SARS-CoV-2, giving an idea about current viral spread status in a certain population.
Reviewer 2 Report
Manuscript ID vaccines-1656563. The study entitled “SARS-CoV-2 anti-nucleocapsid antibody response of mRNA and inactivated virus vaccines compared to unvaccinated individuals” by Dr. Qaqish is a cross-sectional study which aimed to determine and compare the SARS-CoV-2 anti-nucleocapsid antibody levels in personnel vaccinated with Sinopharm vaccine and Pfizer vaccine in comparison with naturally infected patients. The study/control groups comprised COVID-19 confirmed patients (n=99), Pfizer-vaccinated recipients (n=100), and Sinopharm-vaccinated recipients (n=100), while anti-N IgG, anti-N IgM and anti-S IgG were investigated. Main results indicate the presence of a significant positive correlation between anti-N and anti-S among the 3 groups. The study reports interesting findings on the SARS-CoV-2 vaccines, while pointed on the importance of the anti-SARS-CoV-2 vaccination. It therefore will reach a relatively broad audience considering the topic covered, which is indeed important. However, several important improvements should be made for making the work suitable for publication. Also, figures requiring a serious quality improvement. I therefore recommend a major revision. I have few suggestions for improving the manuscript:
Thank you for letting me revise this interesting work
General comments
1. I suggest make more prosaic the aim and the conclusion of the study, thus replacing the list of sub-aims/sub-conclusions with more descriptive sentences
2. Study populations and main ages should be moved from the results to the methods
3. The methods sections “ sample analysis” and “statistical analysis “ are completely lacking in supporting references. Please provides supporting references
4. The quality of all figures should be improved
5. If possible, quantitative results on IgG levels (antibody titres) should be, at least briefly, mentioned in the abstract
6. All rates reported throughout the text of the result section should be in the following form: rate% (positive/total). For instance “50% (25/50)”. Please revise the entire text accordingly. It is the case of lines 155, 156, 172 etc.
7. In case of unsignificant differences, authors should include p>0.05 at the end of the sentences
8. Besides igG and IgM medians, interquartile ranges should also be included in both text and tables
Minor
Lines 46-52 a detailed description of mai anti-SARS-Cov-2 vaccines, including efficacy and safety, is reported here (DOI: 10.3390/v13091687 and DOI: 10.7499/j.issn.1008-8830.2101133). Both references should be included
Line 54 better “spike (S) glycoprotein”
Line 167 the quality of figure 1 should be improved
Line 182 the quality of figure 2 should be improved. Several words are almost unreadable
Line 196 the quality of figure 3 should be improved
Line 258 better “a higher positivity rate (88%)”
Line 263 the meaning of +ve should be included
Author Response
Manuscript ID vaccines-1656563. The study entitled “SARS-CoV-2 anti-nucleocapsid antibody response of mRNA and inactivated virus vaccines compared to unvaccinated individuals” by Dr. Qaqish is a cross-sectional study which aimed to determine and compare the SARS-CoV-2 anti-nucleocapsid antibody levels in personnel vaccinated with Sinopharm vaccine and Pfizer vaccine in comparison with naturally infected patients. The study/control groups comprised COVID-19 confirmed patients (n=99), Pfizer-vaccinated recipients (n=100), and Sinopharm-vaccinated recipients (n=100), while anti-N IgG, anti-N IgM and anti-S IgG were investigated. Main results indicate the presence of a significant positive correlation between anti-N and anti-S among the 3 groups. The study reports interesting findings on the SARS-CoV-2 vaccines, while pointed on the importance of the anti-SARS-CoV-2 vaccination. It therefore will reach a relatively broad audience considering the topic covered, which is indeed important. However, several important improvements should be made for making the work suitable for publication. Also, figures requiring a serious quality improvement. I therefore recommend a major revision. I have few suggestions for improving the manuscript: Thank you for letting me revise this interesting work
General comments
1. I suggest make more prosaic the aim and the conclusion of the study, thus replacing the list of sub-aims/sub-conclusions with more descriptive sentences
Response: thank you. The aims and conclusions were improved as requested as follow:
Aims: This study aims to determine the anti-N Ab response in personnel vaccinated with inactivated SARS-CoV-2 virus vaccine, Sinopharm, in comparison with naturally infected – unvaccinated and Pfizer S protein mRNA vaccinated subjects.
Conclusion: In conclusion, this study shows that the inactivated virus vaccine, Sinopharm, induces an anti-N response that can boost that of natural infection or vice versa. On the other hand, the Pfizer mRNA-based vaccine induces a significantly stronger anti-S Ab response. Many factors could affect anti-S and anti-N antibody levels among vaccinated individuals including age, chronic diseases, and duration of sampling after vaccination. Anti-N Ab detection could be a helpful tool to survey COVID-19 past and current infections among a vaccinated population with mRNA-based vaccines. The role of high levels of anti-N Abs after vaccination, natural infection, or both in development of protective immunity versus ADE related complications needs further investigation.
2. Study populations and main ages should be moved from the results to the methods
Response: Modification done as advised. Thank you.
ADE has been observed in many viral infections, mainly with positive-strand RNA viruses such as dengue virus, influenza virus, human immunodeficiency virus -1 (HIV-1), West Nile virus, MERS and others.
The methods sections “sample analysis” and “statistical analysis “ are completely lacking in supporting references. Please provides supporting references
Response: Modification done as advised. Thank you.
The quality of all figures should be improved
Response: Resolution for all figures have been improved. Thank you.
If possible, quantitative results on IgG levels (antibody titres) should be, at least briefly, mentioned in the abstract
Response: Modification done as advised. Thank you.
All rates reported throughout the text of the result section should be in the following form: rate% (positive/total). For instance “50% (25/50)”. Please revise the entire text accordingly. It is the case of lines 155, 156, 172 etc.
Response: Modification done as advised. Thank you.
In case of unsignificant differences, authors should include p>0.05 at the end of the sentences
Response: P>0.05 has been added to all unsignificant differences listed in the text. Thank you.
Besides igG and IgM medians, interquartile ranges should also be included in both text and tables
Response: Modification done as advised. Thank you.
Minor
Lines 46-52 a detailed description of mai anti-SARS-Cov-2 vaccines, including efficacy and safety, is reported here (DOI: 10.3390/v13091687 and DOI: 10.7499/j.issn.1008-8830.2101133). Both references should be included
Response: Modification done as advised. Thank you.
Line 54 better “spike (S) glycoprotein”
Response: Modification done as advised. Thank you.
Line 167 the quality of figure 1 should be improved
Response: Resolution has been improved. Words have been abbreviated. Thank you.
Line 182 the quality of figure 2 should be improved. Several words are almost unreadable
Response: Resolution has been improved. Words have been abbreviated. Thank you.
Line 196 the quality of figure 3 should be improved
Response: Resolution has been improved. Words have been abbreviated. Thank you.
Line 258 better “a higher positivity rate (88%)”
Response: Modification done as advised. Thank you.
Line 263 the meaning of +ve should be included
Response: Modification done as advised. Thank you.
Round 2
Reviewer 2 Report
The authors have satisfactorily addressed most of my concerns. The ms can be accepted in the present form.
However, these minor corrections can be made before publication:
line 53 "WHO" should be World Health Organization (WHO)
Figures are still difficult to read due to the poor quality
The authors can maintain the percentages (%), only, instead of indicating the (positive/total) in the discussion section